# Molecular EPISTOP, a comprehensive multi-omic analysis of blood from Tuberous Sclerosis Complex infants age birth to two years

We present a comprehensive multi-omic analysis of the EPISTOP prospective clinical trial of early intervention with vigabatrin for pre-symptomatic epilepsy treatment in Tuberous Sclerosis Complex (TSC), in which 93 infants with TSC were followed from birth to age 2 years, seeking biomarkers of epilepsy development. Vigabatrin had profound effects on many metabolites, increasing serum deoxycytidine monophosphate (dCMP) levels 52-fold. Most serum proteins and metabolites, and blood RNA species showed significant change with age. Thirty-nine proteins, metabolites, and genes showed significant differences between age-matched control and TSC infants. Six also showed a progressive difference in expression between control, TSC without epilepsy, and TSC with epilepsy groups. A multivariate approach using enrollment samples identified multiple 3-variable predictors of epilepsy, with the best having a positive predictive value of 0.987. This rich dataset will enable further discovery and analysis of developmental effects, and associations with seizure development in TSC.

Tuberous sclerosis complex (TSC) is a multisystem disorder caused by inactivating mutations in either *TSC1* or *TSC2*[1,2]. Tumors form in many organs in TSC, following the two hit Knudson mechanism, including the heart, brain, kidneys, skin, and lungs[1,2]. However, the predominant morbidity in TSC infants and young children is due to severe and often drug-resistant epilepsy, with infantile spasms and other seizures occurring in 70–90% of TSC infants[3,4]. Neurodevelopmental comorbidities including intellectual disability and autism are also common in TSC, are collectively called TSC-associated neuropsychiatric disorders (TAND)[5,6], and correlate with seizure severity and lack of effective seizure control.

TSC is often diagnosed prenatally, because cardiac rhabdomyomas are a cardinal feature of TSC, and are commonly observed on fetal ultrasound[2]. This situation has enabled consideration of pre-symptomatic antiseizure medication (ASM) for TSC infants, prior to seizure onset, with the goal of prevention or at least diminution of

seizure severity, ideally leading to long-term seizure control and improved neurocognitive outcome[7].

EPISTOP (Long-Term, Prospective Study Evaluating Clinical and Molecular Biomarkers of Epileptogenesis in a Genetic Model of Epilepsy–Tuberous Sclerosis Complex, NCT02098759) was a controlled multicenter study, in which the safety and efficacy of pre-symptomatic ASM treatment with vigabatrin, introduced when focal discharges more than 10% of the time or multifocal discharges were seen on video-EEG, was compared with conventional ASM started after the onset of seizures[8]. In 54 enrolled EPISTOP subjects, epileptiform EEG abnormalities were identified before seizures. The time to the first clinical seizure was significantly longer with presymptomatic than conventional treatment, and at age 24 months, presymptomatic treatment reduced the risk of clinical seizures (odds ratio (OR) = 0.21, $p = 0.032$), drug-resistant epilepsy (OR = 0.23, $p = 0.022$), and the occurrence of infantile spasms (OR = 0, $p < 0.001$)[8].

✉e-mail: dk@rics.bwh.harvard.edu

In addition to the clinical trial, a key part of EPISTOP was to perform extensive multi-omic analyses of subject samples with the goal of identification of biomarkers associated with epilepsy development. Blood samples were collected from subjects at serial time points through age 2 years. Subject DNA samples were analyzed by both whole genome sequencing, and targeted high read-depth sequencing of *TSC1/TSC2* to enable identification of mosaic mutations[9]. Serial blood/serum samples were also analyzed for proteomics, metabolomics, by RNA-Seq, and expression of 45 selected miRNA.

Here we report the results of this analysis, including examination of age-dependent changes and effects of vigabatrin treatment. We focused on examination of associations between analyte levels and risk of seizure development, development of drug-resistant epilepsy, and ongoing seizures at age 2 years. An age-matched control population was also sampled to enable comparison with controls.

## Results

### Sample collection and analyses performed

As part of EPISTOP, blood samples were collected longitudinally from 93 subjects with TSC, from birth until two years of age. Samples were collected at serial time points, dependent upon EEG findings and clinical events during the course of the study: 1) at enrollment; 2) at time of detection of an abnormal epileptiform EEG; 3) at onset of seizures (clinical or electrographic); 4) at 6 or 12 months of age, if no abnormal epileptiform EEG or seizure was detected during the study; 5) at age 24 months (Supplementary Fig. 1a). Control samples from 58 age-matched children with various conditions but without TSC or seizure history were also collected. Serum samples were analyzed for: proteomics by mass spectrometry; 298 preselected water soluble metabolites by mass spectrometry; and 45 miRNA species by Quantitative reverse transcriptase PCR (RT-PCR). RNA was extracted from total blood samples, and analyzed by RNA-Seq (Supplementary Fig. 1b). Whole genome sequencing was performed to permit analysis of single nucleotide polymorphisms (SNPs) potentially associated with epilepsy. *TSC1* and *TSC2* deep sequencing in concert with whole genome sequencing enabled detection of heterozygous and mosaic pathogenic variants[9].

Four hundred and eighty-one protein groups were identified in the proteomics analysis (at false discovery rate (FDR) < 1%). 63,677 RNAs were identified by RNA-Seq (Supplementary Data 1). For the univariate analyses, RNAs and metabolites were retained for further analysis only if non-zero values were seen in ≥ 50% samples; protein groups if seen in ≥ 70%. 20,579 RNAs (13854 protein coding genes, 3221 long noncoding, 2598 of other biotypes, including miRNAs, snRNAs, pseudogenes, etc. and 906 RNAs that could not be assigned to any biotype), 249 metabolites, and 340 protein groups were retained for further analysis.

The intent of our analysis was to identify analytes that were associated with epilepsy outcome in the EPISTOP patient set. We considered many ways of associating analytes with epilepsy outcomes, but first assessed confounding effects that could lead to false associations.

### Confounding effects in single dataset analysis

Several confounding effects were expected and identified. First, sample processing and analysis via mass spectrometry and sequencing was performed in batches, and batch effects were anticipated. Second, vigabatrin (VGB) is known to impact metabolite levels[10,11]. VGB therapy was the main therapeutic intervention in the EPISTOP trial, such that by age 2 years, 89% of all subjects were taking VGB. Third, we anticipated that multiple analytes would vary naturally from newborn to age 2 years, so that accounting for such effects was quite important to avoid confounding by age.

**Batch effects.** Batch effects were observed in the metabolite dataset (Supplementary Fig. 2a, b), but not in any of the other analyses.

Metabolite batch effects were corrected by a Z-score based correction method (see Methods for details).

**Vigabatrin effects on serum metabolites.** VGB therapy clearly affected multiple metabolites, as shown by Principal Component Analysis (PCA) (Fig. 1a). As age also influenced many metabolite levels, we examined VGB effects in detail considering samples only from subjects > 40 weeks of postnatal age, an age range in which age effects were minimal. We compared samples from individuals who were receiving VGB versus those who were not. Since 89% of EPISTOP subjects were taking vigabatrin by age 2 years, we increased the non-VGB group by addition of non TSC samples with similar ages ($n_{max}$ VGB: 72; $n_{max}$ non-VGB: 28). 28 metabolites were found to be significantly different (FDR < 0.05, Wilcoxon rank sum test) (Supplementary Data 2). dCMP, glutathione, glutathione-nega, aminoadipic acid, 4-aminobutyrate, kynurenic acid and 5 others showed a >2.5-fold difference (Fig. 1c–h, Supplementary Data 2). dCMP showed the greatest fold change of all metabolites, being increased 52-fold in subjects taking VGB compared with those that were not on VGB (Fig. 1g).

The VGB effect on selected metabolites was corrected using a Z-score based method (see Methods for details). Following this correction, PCA showed no difference according to VGB use (Fig. 1b), and all metabolites showed a similar distribution in those treated without or with VGB (Fig. 1c–h).

**Developmental effects on analytes.** Multiple serum proteins are known to be highly developmentally regulated, including alpha fetoprotein, which undergoes a decline of approximately 10,000-fold during the weeks after birth[12]. Our longitudinal study design, newborn to age 2 years, enabled the detection of both expected and many unexpected age dependent changes in serum proteins, serum metabolites, and RNA expression. To facilitate analysis of these effects, we divided our population into age tertiles, 0–10 weeks, 11–40 weeks, and > 40 weeks postnatal age.

There was a strong association between age and the first PCA component for both protein groups and metabolites (Fig. 2a, b). However, an association between age and the first two components of the PCA was not readily seen for the RNA data; rather an association was seen with the 3rd, 4th, and 5th components (Fig. 2c). In addition, the majority of metabolites, protein groups, and RNA species showed either a positive or negative correlation with age as a continuous variable (Spearman's rank correlation coefficient, Fig. 2d). One such protein group (tenascin) and one metabolite (ethanolamine) that showed major changes with age are shown in Fig. 2e, f as an example.

Kruskal-Wallis analysis (FDR < 0.05) showed that the majority of analytes had a significant association with age in the 3 group age-tertile comparison: 285 out of 340 (84%) protein groups, 169 out of 249 (69%) metabolites, and 10,506 out of 20,579 (51%) RNAs (Supplementary Fig. 3a–c, Supplementary Data 3).

In the case of protein groups, hemoglobin subunits (alpha, beta, gamma-1 and gamma-2) showed fold changes >38, with higher levels detected in the youngest age group, consistent with hemolysis of red blood cells in blood drawing from infants. As expected, alpha-fetoprotein showed a fold change of approximately 181 with inconsistent detection (close to the detection limit of the mass spectrometer) in the two older age groups (>11 weeks of age) (Supplementary Fig. 4). Overall 147 protein groups showed fold changes > 1.5. Tenascin, C4b-binding protein alpha chain, collagen alpha-1(V) chain, collagen alpha-1(I) chain, complement component C1q receptor, and collagen alpha-1(III) chain all exhibited fold changes >7 between the youngest age group and the middle or oldest age group (Supplementary Fig. 4).

64 metabolites showed median fold changes > 1.5 comparing samples from age <10 weeks to > 40 weeks. The largest fold changes were detected in ethanolamine (fold change = 4.1) and leucine-

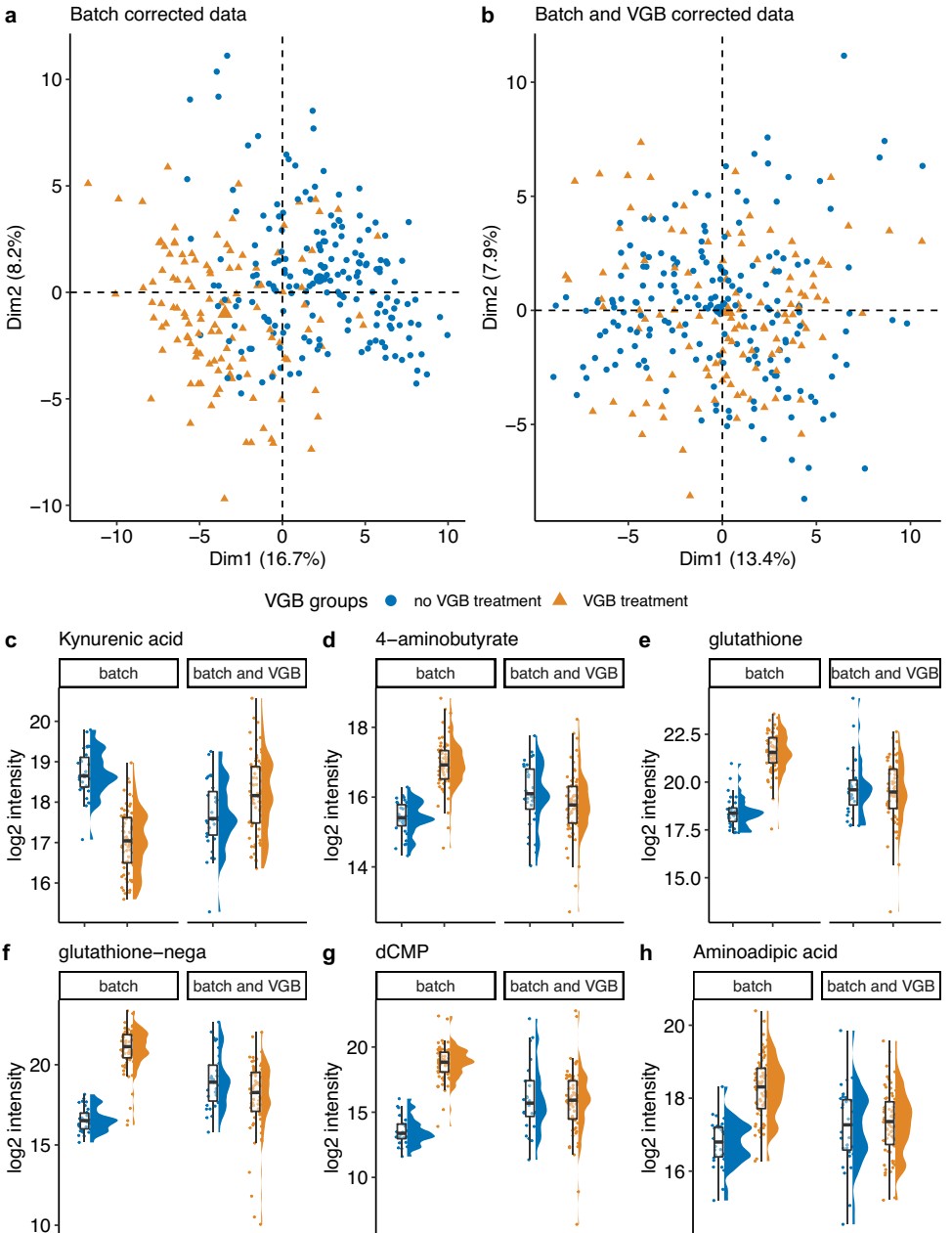

**Fig. 1 | Vigabatrin (VGB) effects on metabolites. a,b** PCA plots of metabolomics demonstrating Vigabatrin (VGB) effect before (**a**) and after correction (**b**) are shown. cdefgh. Correction for effects of VGB. Plots for kynurenic acid (**c**), 4 aminobutyrate (**d**), glutathione (**e**), glutathione-nega (**f**), dCMP (**g**), and aminoadipic acid (**h**) before and after correction for VGB exposure are shown ($n_{max}$ VGB = 72; $n_{max}$ no VGB = 28; FDR < 0.05, Wilcoxon rank sum test). Samples from subjects of age > 40 weeks are shown. The box plot's central line denotes the median, and its lower and upper boundaries indicate the 25th and 75th percentiles of the data, respectively. The whiskers represent the highest and the lowest values no further than 1.5 times the IQR. All data points are shown. Source data are provided as a Source Data file.

isoleucine levels (fold change = 4.4), which decreased with increasing age, and shikimate levels (fold change = 4.7), which increased with increasing age. Four other metabolites exhibited fold changes > 3 (Supplementary Fig. 5).

2512 of 10506 (24%) age regulated RNAs exhibited fold changes > 1.5. Fetal hemoglobin genes *HBG1* and *HBG2* showed reductions of 41.1-fold and 23.4-fold, respectively, an expected developmental event, while *KCNG1* showed a major increase (42.5-fold), and *EVA1A* a major reduction (8.8-fold), comparing the youngest to the oldest age group (Supplementary Fig. 6). XIST, a long non-coding RNA involved in X chromosome inactivation[13], showed marked differences in expression according to sex, as expected (Supplementary Fig. 6). It also appeared

to be differentially expressed by age, due to an imbalance in sex distribution by age.

Hierarchical clustering of age group median Z-scores that were found to be differentially expressed with age in the Kruskal-Wallis analysis (FDR < 0.05) was used to identify similar patterns of expression changes among protein groups, metabolites, and RNA species (Supplementary Fig. 3a–c). Based on visual inspection and tuning, different clusters with similar changes in levels with age were defined for protein groups (6 clusters), metabolites (6), and RNAs (8), respectively (Supplementary Fig. 3a–c, Supplementary Data 3).

To assess if these clusters of age-dependent analyte changes were due to common pathways or developmental processes

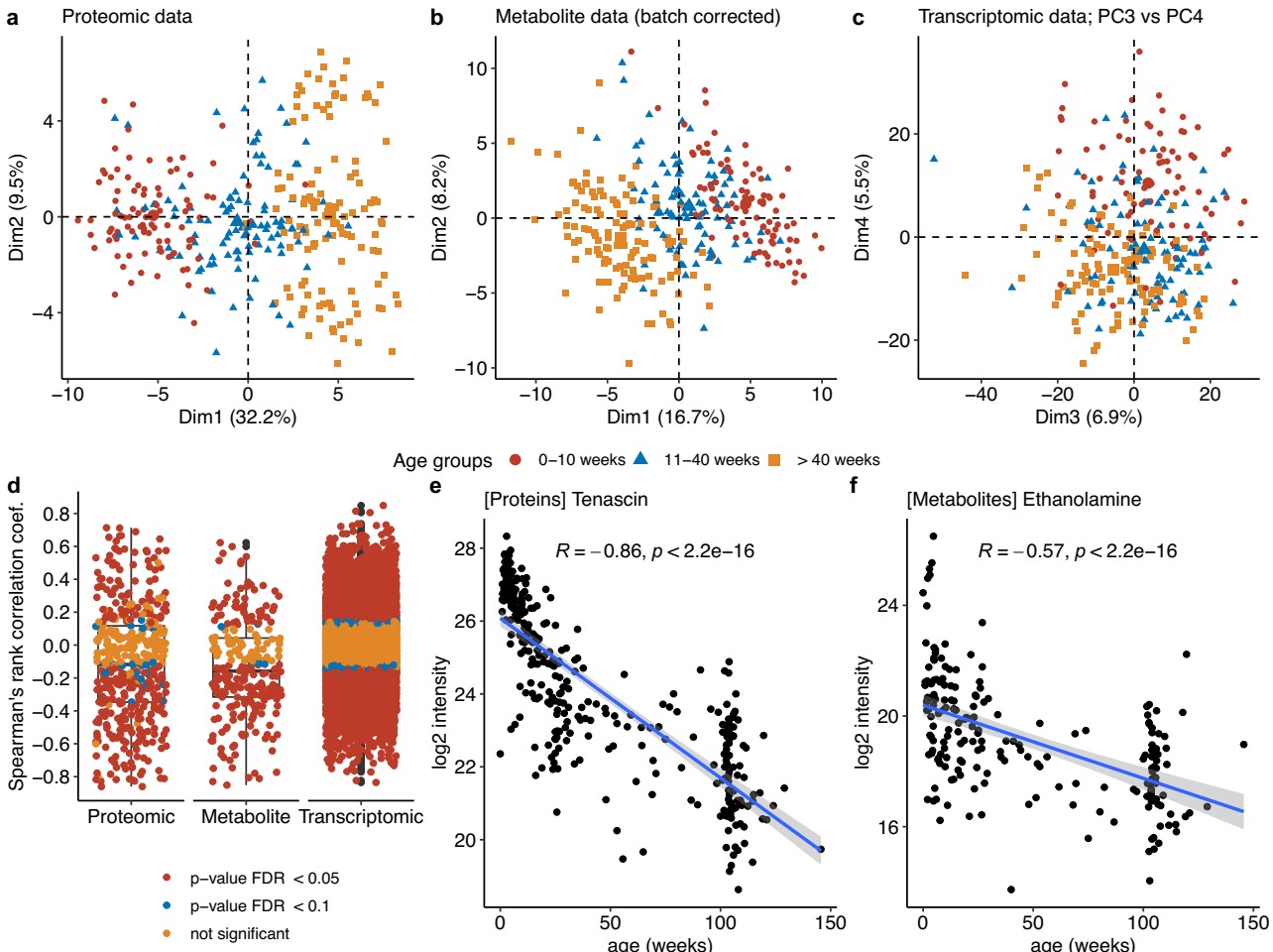

**Fig. 2 | Developmental effects on the proteomic, metabolite and transcriptomic data. a–c** PCA of each analyte set colored by age groups. **d** Spearman's Rank correlation coefficient of each analyte with age, before age correction and colored by $p$ value. Each dot represents one analyte. 187 (75%) metabolites, 286 (62%) protein groups, and 8585 (42%) RNAs showed a significant correlation with age (FDR < 0.05). A two-sided correlation test using Spearman's method was executed, and the results were adjusted for multiple comparisons using the Benjamini-Hochberg procedure. The box's middle line marks the median, its edges represent the 25th and 75th percentiles, and whiskers extend to data points within 1.5*IQR. **ef**. Plots of two individual analytes according to age, prior to correction. Protein group tenascin (**e**), Ethanolamine (**f**) significantly correlated with age ($p < 0.001$). The line is drawn to visualize the trend in the data, representing the linear regression fit, while the surrounding shaded area denotes the 95% confidence intervals (CI). Spearman's Rank correlation coefficients (R) and $p$ values obtained by two-sided Spearman's correlation test are indicated. Source data are provided as a Source Data file.

occurring with age, clusters of analytes were assessed for enrichment in Gene Ontology (GO), KEGG, and Reactome gene sets (Supplementary Data 4a, b). Protein cluster group 2 showed the greatest decline with age, and was enriched for genes involved in extracellular matrix organization and structure in all three gene sets (Supplementary Data 4a). Protein group cluster 3, in which protein levels decreased less strongly over time, was enriched for genes involved in carbohydrate metabolic process by GO gene set analysis (Supplementary Data 4a). Protein group clusters 4 and 6, in which protein levels steadily increased with age, were both enriched for genes involved in complement and coagulation (Supplementary Fig. 3a, Supplementary Data 4a).

None of the metabolite age clusters showed enrichment for any of the Reactome terms which we tested (Supplementary Fig. 3b).

In contrast to the protein and metabolite datasets, there was no strong correlation between gene expression and age. Since it is known that there is significant variation in the fraction of white blood cell types in human blood according to age from birth to age 2 years[14–16], we considered the possibility that some of the variation in expression might be due to variation in the relative numbers of different white blood cell types. However, age appeared to be a minor factor affecting gene expression (Fig. 2c). Nonetheless, CIBERSORT[17] was used to analyze RNAseq data to determine the relative proportions of white blood cell types in each sample. Correlation between the relative proportion of various leukocyte populations was with the first 10 principal components (PC) of the RNA-Seq data (Supplementary Fig. 7a). PC1 of the RNA data showed a positive correlation with the proportion of activated NK cells and regulatory T-cells. The neutrophil proportion correlated strongly negatively with PC2, while B cell and various T cell proportions correlated positively. Several PCs showed some degree of correlation with age, though not PC1 or PC2. The proportion of both monocytes and CD4 naive T-cells declined significantly with age, while the neutrophil fraction more strongly increased with age through two years (Supplementary Fig. 7b).

Differences in the neutrophil vs. lymphocyte fraction with age was reflected in the RNA cluster analysis, which grouped genes into eight clusters. Several of these clusters (1, 2, 3, 6) were enriched for genes related to neutrophil or immune function (Supplementary Data 4b, Supplementary Fig. 3c), consistent with maturation of the immune system with age.

## Association between clinical features and analytes

To enable a search for associations between analytes and clinical events in the EPISTOP cohort, it was necessary to correct for the strong developmental effects on many analytes. Two independent methods were used: 1) a linear mixed model (LMM) approach; and 2) calculating the Z-score for each analyte within each age group, and then Z-score reversal using the standard deviation and median for all samples. Both methods were applied independently, and to be conservative, we report only associations that were significant after FDR correction by each of these two methods. One example of age-based correction by each of the two methods is shown for C4b-binding protein α-chain (Supplementary Fig. 2c).

We sought to identify associations between analytes and a broad set of clinical features, including: I. diagnostic status, TSC vs. control; II. presence of clinical or electrographic seizures vs. no seizures at time of sample collection; III. presence of abnormal epileptiform EEG vs. normal EEG at time of sample collection; IV. drug-resistant epilepsy vs. seizure free or drug controlled epilepsy at age 24 months; V. history of clinical or electrographic seizures vs. no detected seizures during the EPISTOP project using analytes determined at age 24 months (Va) or analytes determined at enrollment (Vb).

(I) Comparisons between non TSC control samples ($n_{max}$ = 52) and TSC samples ($n_{max}$ = 126) without prior treatment and no seizure history, of any age (including samples from individuals that showed abnormal EEGs at sample draw) identified 39 significant associations: three protein groups (catalase, collagen alpha-2(I), Peroxiredoxin-2), and two metabolites (aminoimidazole carboxamide ribonucleotide, leucine-isoleucine), all five higher in TSC c/w control; and 34 RNAs (25 higher in TSC subjects) (FDR < 0.05, fold change > 1.5, Supplementary Data 5, Supplementary Fig. 8). Although these differences were statistically significant, no analyte had levels that could reliably distinguish the control group from the TSC group.

Gene set enrichment analysis of the 25 genes with higher expression in the TSC subjects identified multiple enriched gene sets using each of the GO, KEGG, and Reactome gene sets, many of which related to ribosome structure and function (Supplementary Data 6). In contrast, the 9 genes whose expression was lower in the TSC subjects than controls were enriched in gene sets related to receptor tyrosine kinase and phosphoinositide 3-kinase activity (Supplementary Data 6). These observations may reflect the consequences of haplo-insufficiency for either TSC1 or TSC2 in the blood cells from which the RNA was derived, as complete loss of either TSC1 or TSC2 is known to strongly increase ribosome biogenesis while suppressing upstream receptor tyrosine kinase and phosphoinositide 3-kinase signaling[18].

(II) Comparison of samples drawn at the time of, or shortly after, first seizure occurrence ($n_{max}$ = 86) versus samples from seizure free individuals ($n_{max}$ = 95) showed no significant differences in protein groups, metabolites, or RNAs.

(III) Similarly, comparison of samples drawn at the time of abnormal EEG detection ($n_{max}$ = 45) versus those from subjects with normal EEG ($n_{max}$ = 65) showed no significant differences in any analyte.

(IV) Comparison of samples from EPISTOP subjects with drug-resistant epilepsy at age 24 m ($n_{max}$ = 41) vs. those who were seizure free or drug controlled at age 24 m ($n_{max}$ = 43) led to identification of one protein group, Collagen alpha-1(XI) chain that was significantly higher in those with drug-resistant epilepsy, and one metabolite, Hydroxyphenylacetic acid that was significantly lower in those with drug-resistant epilepsy (Fig. 3). Again, although these differences were statistically significant, neither could distinguish a sample from an individual with drug-resistant epilepsy vs. seizure-free or drug controlled.

(Va) Comparison of samples drawn at age 24 m from EPISTOP subjects with seizure history ($n_{max}$ = 59) vs. subjects without seizure history ($n_{max}$ = 9) vs. subjects without seizure history but pre-

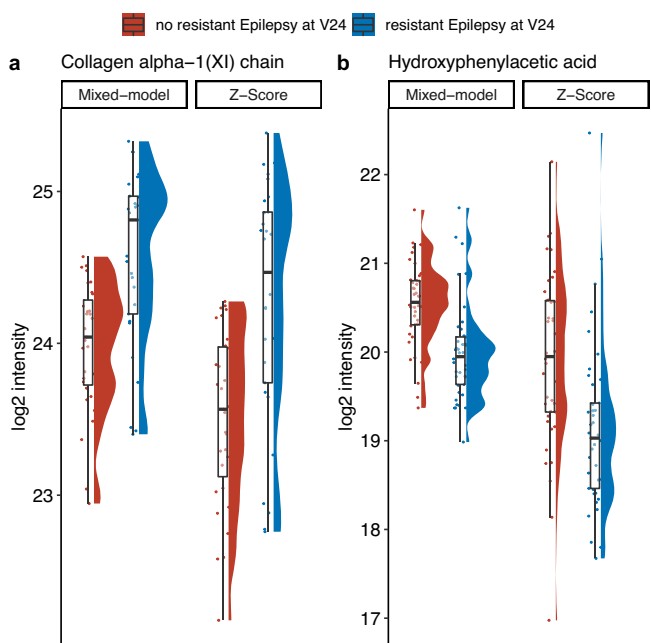

**Fig. 3 | Analytes whose expression at age 24 months was associated with resistant epilepsy.** Collagen alpha-1(XI) chain (**a**) and Hydroxyphenylacetic acid (**b**) are shown, with sample values corrected by two independent methods. These were the only two analytes significantly different in comparison of EPISTOP subjects with refractory epilepsy ($n_{max}$ = 41) vs. those without at age 24 months ($n_{max}$ = 43) (Wilcoxon rank sum test; FDR < 0.05; fold change > 1.5). The median is indicated by the center line of the box, the edges show the 25th and 75th percentiles, whiskers reach to values within 1.5*IQR. All data points are shown. Source data are provided as a Source Data file.

symptomatically treated with VGB ($n_{max}$ = 14) showed no differences for any analyte. Because of the small sample size of groups 2 and 3 we repeated this analysis combining these two groups, and there were still no analytes with significant differences identified.

(Vb) Comparison of analytes determined on samples drawn at enrollment into EPISTOP, comparing subjects who subsequently developed clinical or electrographic seizures vs. subjects who never developed seizures, was analyzed in detail. To enhance the sensitivity of this analysis, we compared three groups: (group 1) non-TSC control; (group 2) TSC subjects who never developed epilepsy during the two year course of the study; and (group 3) TSC subjects who did develop epilepsy. Initially we performed this comparison under the stringent conditions of excluding subjects from group 2 that never developed clinical or electrographic seizures, but had been treated pre-symptomatically with vigabatrin per trial protocol; and excluding subjects from groups 2 and 3 who had an abnormal EEG (but not seizures) at the time of enrollment. With these conditions, there were 34, 9, and 31 ($n_{max}$) samples in groups 1, 2, and 3, respectively. Using the Kruskal-Wallis Rank Sum test, one protein group, Periostin, one metabolite, glucose-6-phosphate, and four RNAs (*ATP5ME, NDUFA1, BHLHA15, CARD16*) were significantly different among the three groups (FDR < 0.05; fold change > 1.5 comparing group 1 to either group 2 or 3; Table 1, Fig. 4).

We repeated this analysis under less stringent conditions, in which group 2 included subjects who never developed seizures but had received pre-symptomatic vigabatrin, per study protocol; and TSC subjects with an abnormal EEG at enrollment were included in both groups 2 and 3. Under these conditions, max group sizes were 34, 24, and 59 respectively. Nine protein groups, five metabolites, and 48 genes showed significant differences in levels (FDR < 0.05; fold change > 1.5, comparing group 1 with either group 2 or 3; Kruskal-Wallis Rank

**Table 1 | Results of statistical comparison to identify potential predictive biomarkers for epilepsy in proteomics, metabolite and RNAseq data**

| molecule_name | alternative name | molecule type | up (+) or down (-) in TSC epilepsy* TSC epilepsy v CTL | up (+) or down (-) in TSC no epilepsy* TSC no epilepsy v CTL | up (+) or down (-) in TSC epilepsy* TSC epilepsy v TSC no epilepsy | KW p value fdr | TSC epilepsy v CTL FC | TSC no epilepsy v CTL FC | TSC epilepsy v TSC no epilepsy FC | sample size control group | sample size TSC no epilepsy | sample size TSC epilepsy |
|---|---|---|---|---|---|---|---|---|---|---|---|---|
| Periostin | Q15063 | protein group | + | | | 6.964E-05 | 1.74 | 1.41 | 1.23 | 34 | 9 | 31 |
| glucose-6-phosphate | | metabolite | + | | | 1.902E-04 | 1.58 | 1.49 | 1.06 | 25 | 8 | 23 |
| ENSG00000125356 | NDUFA1 | RNA | + | | | 1.324E-02 | 1.83 | 1.40 | 1.30 | 24 | 7 | 27 |
| ENSG00000169020 | ATP5ME | RNA | + | | | 1.010E-02 | 2.06 | 1.88 | 1.09 | 24 | 7 | 27 |
| ENSG00000180535 | BHLHA15 | RNA | - | | - | 1.127E-02 | 0.36 | 0.64 | 0.57 | 23 | 7 | 25 |
| ENSG00000204397 | CARD16 | RNA | + | | | 1.010E-02 | 1.67 | 1.34 | 1.25 | 24 | 7 | 27 |

Kruskal-Wallis rank sum test was used to identify significant regulations (FDR <0.05; p value correction using Benjamini-Hochberg procedure; fold change >1.5). Data was previously corrected for batch, VGB treatment (metabolite data) via Z-score correction method and age (protein, metabolite and RNAseq data) using a linear mixed model correction. Only comparisons showing a "+" or "-" in columns marked with "*" were found to be significantly regulated using two-sided Dunn's Multiple Comparison Test (FDR <0.05; fold change >1.5).

Sum test) (Supplementary Data 7). The single protein group, single metabolite, and four RNAs that were identified in the more stringent analysis were also identified in this less stringent analysis.

Eight of nine protein groups were increased in the TSC epilepsy group compared to controls, of which six were also upregulated in the TSC no epilepsy group. These protein groups were enriched in ten GO, nine KEGG, and 11 Reactome gene sets, of diverse types, but showed some consistency with gene sets involving collagen and extracellular matrix structure (Supplementary Data 8a). Only one protein group, Plasma protease C1 inhibitor, was increased in the control group. For the five metabolites, all of which were upregulated in the TSC epilepsy group, there was no significant enrichment in any pathway.

Expression of 16 genes was increased in TSC epilepsy subjects compared to controls, while 26 were decreased in that comparison (Supplementary Data 7). There was apparent enrichment of the 16 genes in multiple gene sets for these genes, which showed limited consistency for electron transport, respiratory chain, and mitochondrial activity; while the 26 genes were enriched in diverse pathways without apparent consistency (Supplementary Data S8b).

### Integrative predictive models of seizure development in TSC

As noted, a primary goal of this analysis of the EPISTOP cohort was to identify predictive biomarkers of epilepsy development. In addition to the above analysis, we performed an integrative approach, using results from all initial on-study samples to develop a predictive classifier for seizure development (Supplementary Fig. 1a). For this analysis, we also included SNP data for SNPs previously associated with epilepsy development in the general population ($n = 86$, see methods, Supplementary Data 11), as well as levels of 45 miRNAs determined for 139 subjects (93 TSC, 46 Control) in 324 serum samples (see Methods, Supplementary Data 10). Due to the large number of analytes (>20,000, considering RNA species), an initial filtering step was performed, in which only those analytes showing an Area-under-the-curve (AUC) > 0.6 for association with epilepsy development in univariate analysis were retained. In addition, a maximum of 30 analytes, those with highest AUC values, from each data type were included. Thus 126 analytes were considered for predictive classifier analysis.

Logistic regression was used to examine all potential combinations of variables, for their ability to predict epilepsy development, using an arbitrary decision boundary for each model. The Matthews Correlation Coefficient (MCC)[19] was used as a metric for the quality of each classifier with the observed data. A maximum of three variables per predictor were considered so that in total ~330,000 1-, 2- and 3-variable models were evaluated. The mean MCC among all classifiers was 0.4; 295 classifiers had MCC values > 0.70; 5 had values >0.80. A permutation test was used to assess the quality of the predictor using randomized data (Supplementary Fig. 9). Based on the threshold defined by the permutation test, 100 models were considered statistically significant; these had mean MCC values between 0.733 and 0.873. The ten best classifiers and their evaluation metrics (Table 2), and the complete list of statistically significant classifiers (Supplementary Data 9A) are shown. Twenty-two analytes/characteristics appeared in > 4 of the top 100 classifiers (Supplementary Data 9B), including a SNP on chromosome 16 seen in 55 classifiers, and serum carnitine levels seen in 26 classifiers, for which lower levels at study entry were associated with seizure risk (Table 3).

The strongest predictor was a 3-variable classifier consisting of miR-130a-3p, *CECR7* and *RADX* (Fig. 5a). Seizure-free EPISTOP subjects had lower normalized expression of miR-130a-3p (<5 M) and *RADX* expression (<0.5 CPM), compared to those with seizures. *CECR7* had a more minor effect on seizure prediction, although no patient without seizure had a *CECR7* level <0.5 CPM. This classifier had a mean test MCC of 0.873, and strong positive predictive power (PPV = 0.987).

Using the two variables most frequently seen in the classifiers, SNP rs1046276 T/C, and serum carnitine, a 2-variable classifier was

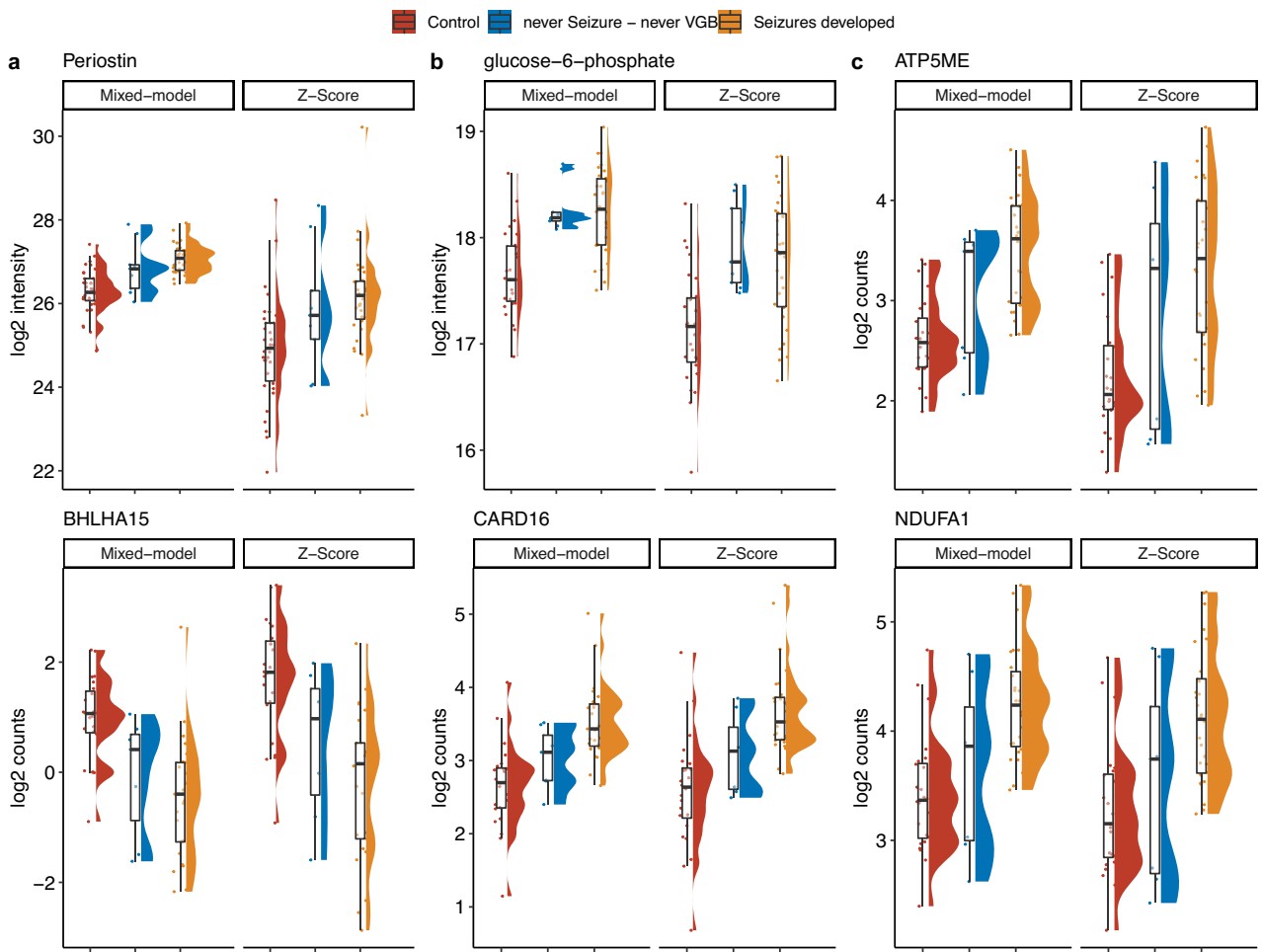

**Fig. 4 | Analytes associated with epilepsy and/or TSC. a**, protein groups; **b**, metabolites; and **c**, mRNA species that were significantly different (Kruskal-Wallis Rank Sum test (FDR < 0.05; fold change > 1.5) in the comparison of group 1 with either group 2 or group 3 of three groups: (1) non-TSC control ($n_{max}$ = 34); (2) TSC subjects who never developed epilepsy during the two year course of the study ($n_{max}$ = 9); and (3) TSC subjects who did develop epilepsy ($n_{max}$ = 31). The median is indicated by the center line of the box, the edges show the 25th and 75th percentiles, whiskers reach to values within 1.5*IQR. All data points are shown. Source data are provided as a Source Data file.

**Table 2 | Top 10 predictive classifiers for seizure development based on the Matthew Correlation Coefficient (MCC) metric**

| Features | $n$ total | LR threshold | MCC test mean | MMCE test mean | PPV test mean | NPV test mean | $p$-value |
|---|---|---|---|---|---|---|---|
| hsa-miR-130a-3p + CECR7 + RADX | 57 | 0.321 | 0.873 | 0.039 | 0.987 | | 0.019 |
| hsa-miR-130a-3p + RADX + rs951997 A/G chr2 223567016 | 56 | 0.285 | 0.839 | 0.046 | 0.973 | 0.892 | 0.025 |
| hsa-miR-130a-3p + RADX + rs16944 A/G chr2 113594867 | 56 | 0.205 | 0.836 | 0.045 | 0.968 | 0.918 | 0.025 |
| PARM1 + rs1046276 T/C chr16 30914626 + rs951997 A/G chr2 223567016 | 61 | 0.632 | 0.807 | 0.053 | 0.973 | 0.862 | 0.031 |
| MIR4539 + rs1046276 T/C chr16 30914626 + rs211037 C/T chr5 161528280 | 61 | 0.423 | 0.802 | 0.051 | 0.965 | 0.9 | 0.032 |
| CORO2B + RADX + rs16944 A/G chr2 113594867 | 61 | 0.711 | 0.796 | 0.052 | 0.971 | | 0.033 |
| ENSG00000266408 + rs1046276 T/C chr16 30914626 + r211037 C/T chr5 161528280 | 61 | 0.487 | 0.794 | 0.051 | 0.961 | 0.914 | 0.034 |
| LINC00689 + LINC02576 + rs211037 C/T chr5 161528280 | 61 | 0.317 | 0.791 | 0.054 | 0.967 | | 0.034 |
| carnitine + rs1046276 T/C chr16 30914626 | 62 | 0.645 | 0.79 | 0.069 | 0.968 | 0.821 | 0.035 |
| TSC1 + LINC00689 + LINC02576 | 62 | 0.539 | 0.79 | 0.064 | 0.96 | | 0.034 |

Note: Using only the initial on-study samples; $n_{max}$ Seizures = 54; $n_{max}$ no seizures = 11. The p value was determined through a one-sided permutation test.

generated. Nine of 10 patients who were seizure-free were homozygous for C at rs1046276 and had high normalized levels of carnitine (>1). The mean test MCC for this classifier was 0.79, with mean PPV = 0.97, and mean negative predictive value (NPV) of 0.82 (Fig. 5b).

## Discussion

Prior studies examining multiomic changes in children have focused primarily on the first week of life[20–22]. An elegant integrated study on infants from The Gambia and Papua New Guinea demonstrated several

**Table 3 | Most common analytes in the top 100 classification models (n_max Seizures = 54; n_max no seizures = 11)**

| Variable | Appears in N models | up (+) or down (-) in seizures group | Seizures (median (IQR)) | No seizures (median (IQR)) | Mean MCC | Mean MMCE | Mean PPV | Mean NPV |
|---|---|---|---|---|---|---|---|---|
| rs1046276 T/C chr16 30914626 | 55 | | hm: 33% / ht: 67% | hm: 90% / ht: 10% | 0.75 | 0.073 | 0.959 | 0.82 |
| Carnitine | 26 | — | 0.95 (0.89–1.03) | 1.05 (1.03–1.11) | 0.748 | 0.079 | 0.956 | 0.818 |
| RADX (ENSG00000147231) | 14 | + | 0.33 (0.11–0.55) | 0.02 (0–0.14) | 0.774 | 0.068 | 0.962 | 0.836 |
| PARM1 (ENSG00000169116) | 12 | — | 1.54 (0.67–2.44) | 3.75 (2.88–4.61) | 0.75 | 0.071 | 0.967 | 0.797 |
| IGLV3-24 (ENSG00000253822) | 12 | — | 0.02 (0–0.14) | 0.24 (0.06–0.44) | 0.742 | 0.063 | 0.949 | 0.871 |
| CAV2 (ENSG00000105971) | 10 | + | 0.65 (0.35–1.18) | 0.14 (0.06–0.24) | 0.752 | 0.069 | 0.943 | 0.909 |
| rs211037 C/T chr5 161528280 | 9 | | hm: 41% / ht: 59% | hm: 80% / ht: 20% | 0.764 | 0.06 | 0.956 | 0.892 |
| IGKV1-9 (ENSG00000241755) | 8 | — | 2.64 (1.47–4.14) | 6.04 (3.91–7) | 0.749 | 0.07 | 0.956 | 0.828 |
| GPM6A (ENSG00000150625) | 7 | — | 2.23 (1.05–3.13) | 5 (4.05–6.31) | 0.741 | 0.073 | 0.96 | 0.8 |
| ENSG00000254859 | 7 | — | 0.57 (0.2–1.33) | 1.6 (1.17–2.07) | 0.755 | 0.074 | 0.964 | 0.806 |
| hsa-miR-130a-3p | 7 | + | 2539342.26 (1278752.8–5076227.22) | 1545365.73 (1293279–1811184.58) | 0.789 | 0.062 | 0.969 | 0.835 |
| rs6872795 A/G chr5 35677385 | 7 | | hm: 46% / ht: 54% | hm: 80% / ht: 20% | 0.751 | 0.068 | 0.959 | 0.82 |

pathways that were operative during development from newborn to age 1 week, particularly interferon signaling, the complement cascade, and neutrophil activity[20]. Genes with decreasing expression (in leukocytes) during the first week of life were enriched for those involved in cellular responses to stress, detoxification of reactive oxygen, and heme biosynthesis and iron uptake. Genes with increasing expression during the first week were involved in interferon signaling, Toll-like-receptor (TLR), negative regulation of Retinoic Acid Inducible Gene I (RIG-I) and complement activation[20]. Metabolomic differences detected during this age interval involved pathways related to plasma steroids and carbohydrates[20].

We have significantly extended these studies, by expanding the age range examined from birth to age 2 years. In fact, these prior studies examined a perinatal time interval that was not available to us for the vast majority of our samples.

Although many medications undoubtedly have important effects on metabolite levels, our findings on the effects of vigabatrin on serum metabolites in infants and children are striking (Fig. 1, Supplementary Data 2). Twenty-eight of 249 (11%) of metabolites showed a significant change in subjects on vigabatrin; 11 showed a >2.5-fold change; and dCMP showed a huge 52-fold increase in EPISTOP subjects on VGB (Fig. 1). Prior studies have focused on breakdown products of VGB and effects on brain metabolites[10,11]. VGB is well-recognized as the most effective ASM for treatment of TSC-associated seizures[23,24]. Hence it was selected as the intervention in this trial to attempt to prevent epilepsy. However, VGB is associated with a wide variety of side-effects, the most feared of which is vigabatrin-associated visual field loss[25–27]. The frequency and significance of this visual field loss is controversial[28,29], but it is reported to occur as a characteristic pattern of progressive centripetal loss. The possibility that one of these metabolic derangements induced by VGB contributes to either its therapeutic benefit or vigabatrin-associated visual field loss deserves further investigation.

We found that a striking number of all analytes examined showed significant changes in levels going from newborn to age 2 years (Fig. 2d, e, f). Dividing the EPISTOP and control population into three tertiles (postnatal age ≤ 10 weeks, 11-40 weeks, and > 40 weeks), the majority of protein groups, metabolites, and expressed genes showed significant changes with age. Hierarchical clustering of protein groups, metabolites, and expressed genes indicated that these expression changes occurred in similar patterns within these analyte types (Supplementary Fig. 3), and showed enrichment in multiple gene sets, suggesting that underlying global developmental processes were driving these changes in expression for each analyte class.

Changes in leukocyte percent and total number are known to occur during the age of birth to two years[14–16]. Total leukocyte counts are higher in newborns, with relative high levels of monocytes and lymphocytes. Proportions of monocytes and lymphocytes decline through age 2 years, and the relative proportion of neutrophils increases during this interval. We observed these same changes with age in the EPISTOP cohort, with a specific decline in naïve CD4 + T cells and monocytes, as assessed using CIBERSORT and the RNA-Seq data (Supplementary Fig. 7a, b). Principal component analysis of the expression data showed many strong associations between leukocyte subsets determined by CIBERSORT and the different principal components (Supplementary Fig. 7a), likely reflecting variability in leukocyte fractions in different samples due to variable clinical conditions (recent infections, vaccinations, environmental exposures, diet, etc.).

Comparisons between non-TSC controls and TSC samples at a range of ages identified three protein groups, two metabolites, and 34 gene RNAs whose levels were significantly different between these two populations. As noted, although statistically significant, no analyte showed differences that could reliably distinguish a sample from one group versus the other. Gene set enrichment analysis of the 9 genes whose expression was lower in the TSC subjects than the

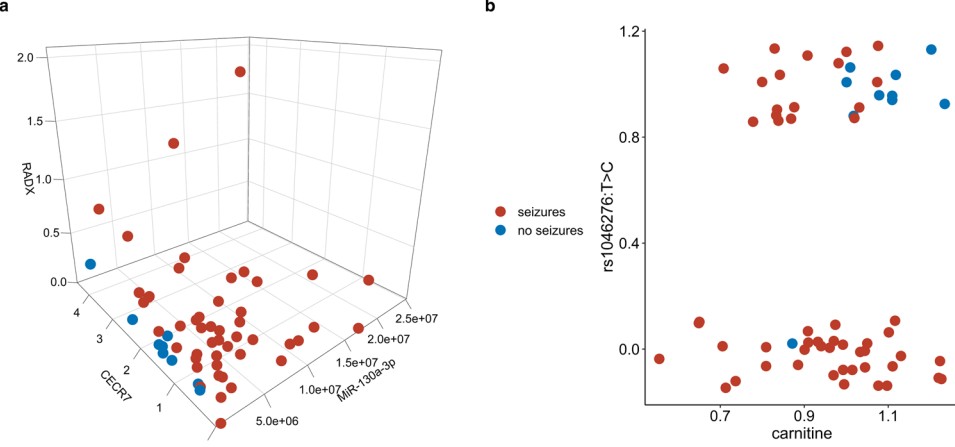

**Fig. 5 | Results of integrative predictive epilepsy analysis. a** 3D plot of three-variable model consisting of miR-130a-3p, *CECR7* and *RADX* (mean test MCC = 0.873). Low level of miR-130a-3p (<5M normalized Unit) and RADX expression (<0.5 CPM) corresponds with seizure-free status. **b** The two-variable model consisting of carnitine and rs1046276 T/C (mean test MCC = 0.79). Homozygote for C at position rs1046276 T/C in combination with high levels of carnitine is associated with remaining seizure free at age 2 years. Source data are provided as a Source Data file.

controls showed that genes were enriched for receptor tyrosine kinase and phosphoinositide 3-kinase functions. Since the TSC subjects would have a single allele loss of TSC1 or TSC2 in general, this may fit with a reduction in the activity of these pathways due to a negative feedback loop[18,30].

It is somewhat disappointing to note the lack of significant differences in any analyte considering samples drawn at time of seizure or aEEG compared to those who were seizure-free. Although alterations in any of the analyte classes might have been predicted, metabolomic differences reflecting effects of seizures or aEEG were anticipated but were not seen. This may be due in part to timing issues such that blood samples for analysis were not always drawn on the same day as diagnosis of seizure/aEEG, due to central review and interpretation of EEGs, delaying recognition of seizures/aEEG in some instances, as well as blood-brain barrier effects.

Collagen alpha-1(XI) protein group was significantly higher in those with resistant epilepsy at age 2 years; while hydroxyphenylacetic acid was significantly lower in those with resistant epilepsy at that age. Col11a1 is a marker of early-born pyramidal neurons[6], and it is possible that its serum marker reflects increased pyramidal neuron development in TSC patients with resistant epilepsy. Hydroxyphenylacetic acid levels in cerebrospinal fluid have been reported to be lower in both schizophrenic and epileptic patients[31], potentially explaining its reduced levels in TSC patients with drug-resistant epilepsy.

A primary goal of this analysis was to identify biomarkers that could be assessed at birth, and might be associated with seizure development in TSC infants. A reliable biomarker of this kind could be developed into a test which might be applied to TSC newborns, to help to identify those TSC infants at high risk of seizure development, and in whom pre-symptomatic use of vigabatrin or other interventions might be taken. To enhance the sensitivity of this analysis, we compared three different groups: control, TSC no epilepsy, and TSC epilepsy groups, finding 6 analytes that were significantly different (Table 1, Fig. 4). Periostin levels were higher in the TSC epilepsy group than the other two groups, and it is known as a biomarker of asthma severity[32], though not previously identified in the context of TSC. *ATP5ME* and *NDUFA1* showed higher expression in blood RNA of the TSC epilepsy group compared with the other two groups, and encode mitochondrial components important for mitochondrial energy production. Their increase in both TSC without seizures and more so in TSC with seizures (Fig. 4) suggests the possibility that TSC alone and with seizures enhances a need for mitochondrial energy production in

blood cells. The other two genes associated with TSC and epilepsy were *BHLHA15*, a basic-loop-helix transcription factor and *CARD16*, an inhibitor of apoptosis; both of uncertain relevance to TSC pathogenesis. However, none of these six markers absolutely discriminated between those developing seizures and those who did not.

A second effort to identify analyte differences that were associated with seizure development was a multivariate approach using logistic regression, resulting in many (100) 3-variable predictors of seizure development (Table 2, Supplementary Data 9), with the best having a mean test MCC of 0.873 with PPV of 0.987 (Fig. 5a).

We recognize the limitation of the size of this patient dataset, in that only 93 subjects with TSC were studied, and only 65 were available for seizure association analysis, since many had received pre-symptomatic vigabatrin. In addition, additional follow-up beyond 2 years will be valuable, and is ongoing at this time. Hence we view all of these observations as exploratory and tentative, and hope that a replication population may be studied for validation in the near future.

## Methods

Full details on the EPISTOP subjects and the prospective clinical trial in which they participated have been reported[8]. Briefly, this study was part of the EPISTOP project (Long-Term, Prospective Study Evaluating Clinical and Molecular Biomarkers of Epileptogenesis in a Genetic Model of Epilepsy–Tuberous Sclerosis Complex, ClinicalTrials.gov NCT02098759). It was carried out from March 2014 to October 2018 at 9 sites in Europe and 1 site in Australia. The study was approved by local ethics committees at all study sites, and caregivers of all participants signed informed consent before enrollment. It adhered to the International Conference on Harmonization Guidelines for Good Clinical Practice and the Declaration of Helsinki.

Blood samples were collected from 93 TSC EPISTOP subjects, 297 different samples in total, and 58 control hospitalized infants without TSC from whom a single sample was collected. Serial samples were collected from TSC subjects in a manner depicted in Supplementary Fig. 1. Blood samples were collected at recruitment into the study, age range birth to age four months. EPISTOP TSC subjects were monitored intensively as part of this trial, including serial EEG monitoring for abnormal activity. When an abnormal EEG was detected a second blood sample was drawn. Another blood sample was collected when either a clinical seizure or subclinical seizure was identified. If no seizure was observed during the first 6 months to 12 months, a blood sample was drawn at one of these timepoints. All TSC subjects had a

last blood sample drawn at age 24 months, the end of the study. Due to delays in recognition of abnormal EEGs or caregiver's report of seizures, those samples were often collected 1–2 weeks following those clinical events. The metadata available for each sample included but was not limited to: TSC mutation type, family history of TSC, epilepsy in the family, presence of seizures (clinical or subclinical), presence of tubers, presence of radial migration lines, white matter abnormalities, gender, grouping (observational or randomized study arm), treatment type (conservative or preventative), infantile spasms, drug-resistant epilepsy (defined as either: 1) ongoing seizures despite the administration of 2 ASM with appropriate doses; or 2) special treatment (epilepsy surgery, ketogenic diet, ACTH)), treatment sequence, vigabatrin treatment, and presence of autism spectrum disorder (ASD) and/or developmental delay (DD) by psychological examination.

Aliquots of serum samples were processed for proteomic analysis as follows. Samples were thawed and centrifuged for 10 min at 16,000x $g$ at 4 °C. 320 μL MARS buffer A (Agilent Technologies) were added to a Cellulose Acetate spin filter (0.22 μm, Agilent Technologies). 80 μL of the centrifuged serum sample was transferred to the MARS buffer A filled spin filters and the mixture centrifuged for 3 min at 16,000x $g$ at 4 °C. The filtrate was transferred into a HPLC vial and stored at 4 °C until affinity chromatography depletion.

Affinity chromatography was performed using MARS 14 depletion columns (Agilent Technologies; Cat: 5188-6558) on an 1100 Agilent HPLC system following the manufacturer's instructions. Flowthrough and bound fraction were collected and precipitation of proteins performed overnight at 4 °C using Trichloro-acetic acid (TCA). Precipitate was washed twice using 90% Acetone (v/v), air dried and suspended in 50 μL 8 M Urea and frozen at −20 °C.

Protein concentration was measured using Bradford assay. 10 μg of protein were reduced with 4.4 mM tris(2-carboxyethyl)phosphine (TCEP) at room temperature and afterwards alkylated for 30 min in the dark with 5.9 mM iodacetamide. 2 μL of Lys-C (0.1 μg/μL; Wako Chemicals) was added to the mixture and a predigestion performed for 2 h at 37 °C. The mixture was then diluted with 56 μL 50 mM triethylammonium bicarbonate and trypsin digestion started using 2.5 μL trypsin (0.04 μg/μL; Promega) at 37 °C overnight. 2.5 μL fresh trypsin (0.04 μg/μL) was added the next day. After further 4 h of incubation at 37 °C, digested samples were frozen at −20 °C until acidification with 5 μL 20% (v/v) formic acid and LCMS-analysis.

2 μg of serum peptides were analyzed via LCMS on a Dionex 3000 Ultimate HPLC System (Thermo Scientific) coupled to an Orbitrap XL mass spectrometer (Thermo Scientific) in a data-dependent acquisition mode. Injected peptides were washed on a temperature-controlled (5 °C) μ-precolumn (C18 PepMap 100, 5 μm, 100 Å; Thermo Scientific) with a flow rate of 25 μL/min using a loading buffer composition 0.5% formic acid and 0.5% acetonitrile in ddH2O. After 4 min peptides were eluted over an analytical reversed-phase column (ReproSil-Pur 120, C18-AQ, 3 μm 500 × 0.075 mm; Dr. Maisch GmbH) at 450 nL/min (Buffer A: 0.1% formic acid in ddH2O; Buffer B: 0.1% formic acid in acetonitrile) using the following gradient: 1–26% B in 149 min, 26% to 46% B in 74 min, 46–90% B in 1 min, holding 90% B for 11 min, 90–1% B in 0.5 min and holding 1% B for 11.5 min. To reduce the amount of carryover between each sample, an injection of 10 μL 80% acetonitrile and a 30 minute wash run was performed using the following gradient: 1% to 90% B in11 min, holding 90% B for 5 min, 90–1% B in 1 min, holding at 1% B 9 min.

The Orbitrap XL was run in positive mode in a data-dependent acquisition manner. A full scan (375–1600 m/z) in the orbitrap analyzer at a resolution of 30,000 (maximum injection time 100 ms, AGC target value $10^6$) was followed by up to 10 MSMS in the ion trap analyzer (minimal intensity threshold 10,000 counts, Isolation width 2.5 m/z, normalized collision energy CID 2.5 eV, maximum ion injection time 25 ms, AGC target value $10^4$). Charge state screening was enabled and only precursors with a charge state of 2 and higher were allowed for data dependent fragmentation. Additionally dynamic exclusion was enabled with a repeat count of 2 in 30 s before exclusion of the precursor (± 5 ppm) for 900 sec. Early expiration was allowed to remove precursors from the exclusion list after an expiration count of 7 full scans with a signal-to-noise ratio of lower than 3.

To assure quality over the processing pipeline in each batch of processed samples a serum pool derived from various female and male subjects with and without diagnosed TSC between the age of 3 months and 3 years of age was also processed as described above. These control samples were measured in a randomized manner in between every 5–10 samples.

Raw files were processed using MaxQuant software (version 1.6.1.0)[33,34] with the following settings: carbamidomethyl of cysteine as fixed modification; variable modification of methionine oxidation and deamidation of glutamine and asparagine; match between runs was enabled with a matching window of 2 min and an alignment window of 20 min; peptide spectrum match FDR, as well as peptide and protein FDR were set at 1%, LFQ with a minimum ratio count of 1. Peptide spectrum matches were analyzed against the MaxQuant contamination database and a fasta file consisting of only reviewed entries from Homo sapiens (Uniprot, downloaded 3rd September 2018). Downstream processing was performed in Excel, Perseus[35] software and R.

Blood samples for transcriptomic analysis and processing were collected and stored in PAXgene tubes. The total RNA was isolated using the PAXgene blood RNA kit in accordance with the manufacturer's guidelines (Qiagen). Library preparation and sequencing was completed at GenomeScan (Leiden, the Netherlands). Sequencing libraries were prepared using the NEBNext Ultra Directional RNA Library prep Kit for Illumina (New England Biolabs) in accordance with the manufacturer's guidelines. Globin mRNA was removed using Invitrogen's Globin clear kit (Waltham) and mRNA was isolated from total RNA using oligo-dT magnetic beads. After fragmentation of the mRNA cDNA synthesis was performed. Sequencing adapters were then ligated to the cDNA fragments, and the resultant product was subject to PCR amplification. Clustering and DNA sequencing was performed using the Illumina HiSeq 4000. All samples were subjected to paired-end sequencing with a read length of 151 nucleotides.

Processing of the raw RNA-seq was carried out by GenomeScan using custom in-house scripts. Read quality was assessed using an in-house QC v1 tool and Trimmomatic v0.30 was used to filter out reads of low quality and any contaminating adapter sequencers[36]. The reads that passed the quality control steps were then aligned to the homo sapiens reference genome (GRCh37) using Tophat v 2.0.14[37]. Next, featureCounts from the Subread package was used to calculate the number of reads that aligned to each gene producing an unnormalized gene count matrix[38]. The unnormalized count matrix was then normalized for library size using the trimmed mean of M-values (TMM) and then converted to count per million values using edgeR[39]. The median number of reads per sample that aligned to known genes was 45 (IQR: 36–74) million reads. Samples with less than 15 million reads aligning to known genes, and one low-quality sample, were discarded.

Deconvolution of the bulk RNA-sequencing was performed using CIBERSORTx[17]. The CIBERSORTx module "Cell Fractions" was used to enumerate the proportions of distinct cell subpopulations in the bulk RNA-Seq profile. This is performed by providing a matrix that has signature genes derived from either single-cell transcriptomes or sorted cell populations. For the deconvolution of the EPISTOP RNA-Seq the publicly available LM22 signature matrix was used. LM22 is a signature genes file consisting of 547 genes that accurately distinguish 22 mature human hematopoietic populations and activation states, including seven T cell types, naïve and memory B cells, plasma cells, NK cells, and myeloid subsets.

miRNA PCR array analysis was performed on 45 miRNAs (Supplementary Data 1, 10). The 45 miRNAs were chosen based on small RNA-seq data from a subset of the serum samples, or possible relevance to

epilepsy, seizures or cognitive functioning according to literature[40–44]. RNA was isolated from serum samples using the miRNeasy Serum/Plasma kit (Qiagen). cDNA was synthesized using the miRCURY LNA SYBR Green PCR Kit and miRNA levels were determined using miRCURY LNA miRNA Custom PCR panels (Qiagen). Plates were run on a Roche LightCycler 480 thermocycler (Roche Applied Science). Quantification of data was performed using the computer program LinRegPCR[45] in which linear regression on the Log (fluorescence) per cycle number data is applied to determine the amplification efficiency per sample. Inter-plate variability was normalized using the inter-plate calibrator assay (UniSp3 IPC) according to the manufacturers' guidelines. The starting concentration of each product was divided by the starting concentrations of two 'spike-in' miRNAs (cel-miR39 and UniSp6) that were added during RNA isolation and cDNA synthesis, respectively, to protect against technical variation and to normalize expression patterns.

Targeted metabolite analysis of serum samples was performed as described[46]. Briefly, 100 μL of isolated serum was stored at −80 °C until processing; samples were thawed at 4 °C and spiked with 10 μM alanine D3 and succinic acid D4 and then centrifuged for 10 min at 14,000x $g$. The supernatant was then added to −80 °C cooled methanol to result in an 80% (vol/vol) methanol solution, gently shaken, and incubated overnight at −80 °C. Samples were then centrifuged for 10 min at 14,000x $g$ (twice if separate liquid phases were observed). Supernatant was collected and stored at −80 °C before being lyophilized under no heat for 3–4 h. Pellets were then suspended in 20 μL HPLC grade water, centrifuged at 14,000x g for 10 min, and 5 μL of supernatant submitted for mass spectrometry analysis coupled to liquid chromatography.

Samples were analyzed using a hybrid 6500 QTRAP triple quadrupole mass spectrometer (AB/SCIEX) coupled to a Prominence UFLC HPLC system (Shimadzu) via selected reaction monitoring (SRM) of a total of 298 endogenous water-soluble metabolites for steady-state analyses of samples. Some metabolites were targeted in both positive and negative ion mode for a total of 307 SRM transitions using positive/negative ion polarity switching. ESI voltage was +4950 V in positive ion mode and −4500 V in negative ion mode. The dwell time was 3 ms per SRM transition and the total cycle time was 1.55 seconds. Approximately 10–14 data points were acquired per detected metabolite. Samples were delivered to the mass spectrometer via hydrophilic interaction chromatography (HILIC) using a 4.6 mm i.d x 10 cm Amide XBridge column (Waters) at 400 μL/min. Gradients were run starting from 85% buffer B (HPLC grade acetonitrile) to 42% B from 0 to 5 min; 42% B to 0% B from 5 to 16 min; 0% B was held from 16 to 24 min; 0% B to 85% B from 24 to 25 min; 85% B was held for 7 min to re-equilibrate the column. Buffer A was composed of 20 mM ammonium hydroxide/20 mM ammonium acetate (pH = 9.0) in 95:5 water:acetonitrile. Peak areas from the total ion current for each metabolite SRM transition were integrated using MultiQuant v3.0 software (AB/SCIEX) and used as the basis for quantitative analysis. All metabolites with >50% missing values across all 357 samples were dropped from analysis, leaving a total of 249 metabolites for further consideration. Repeated measurements for some of the control samples were performed as a quality check between acquisition batches.

To remove observed batch and Vigabatrin effects in the metabolite data set a correction was applied. Individual samples were annotated with their respective batch number and whether the subject was on vigabatrin. A Z-score correction of samples from individual batches was performed on log2-transformed peak area values followed by un-Z-scoring using the standard deviation and mean of all samples prior to Z-scoring. VGB correction was then performed separately.

Age correction was applied using two different Methods: A Z-score un-Z-score method, and a Linear Mixed Models approach. For the Z-score un-Z-score method, samples were divided into three tertiles of age groups: 0–10 weeks, 11–40 weeks, >40 weeks of age. Z-scoring of log2 transformed intensity/integrated peak area/count values was performed in individual age groups and followed by un-Z-

scoring using the mean and standard deviation of all applicable samples prior to Z-scoring. Linear Mixed Models (LMM) was also used, since there were repeated measurements on the same subject over time. The LMM approach permits characterization of time trends both within and between subjects. The formula for the LMM model for the i-th subject is as follow:

$$Y_i = X_i\,\beta + Z_i\,u_i + \varepsilon_i$$

Where:

$Y_i$ - vector of responses, here – measured intensity/peak area/counts

$\beta$ - fixed effects (time-variant), constant across individuals; here – the age at the moment of sampling,

$u_i$ - random effect (time-invariant), grouping factor; here – patient's code.

We estimated the random and fixed effects using lme4 and lmerTest packages in R. We have treated the patient's code as a random effect and the age as a fixed effect with the application of the following formula:

$$lmerTest :: lmer(intensity \sim sample\_age\_w + (1|code)) \qquad (1)$$

By transforming the equation, we can derive a formula for intensity correction, depending on the sampling time and accounting for inter-patient variability:

$$i_{corr} = i - (age)_t * a - b_{ID} \qquad (2)$$

where:

i - intensity

t - time of sampling

a – fixed effect for each week of age

b - model intercept (random effect for each patient)

ID - patient's ID

The assessment of developmental patterns in analytes involved several steps. The first step was to apply the Kruskal-Wallis Rank Sum test, alongside Dunn's post hoc test and Benjamini-Hochberg correction for multiple comparisons for each analyte and age category. Then for analytes with statistically significant differences among age groups, normalization was performed on all values for each molecule using Z-score standardization. To group molecules demonstrating similar patterns, the medians of standardized values were used as input for hierarchical clustering analysis. The process of hierarchical clustering using the Complete linkage method was executed on a dissimilarity matrix derived from Euclidean distance calculations.

For two-group comparisons, first, we verified the t-test assumptions: data normality using the Shapiro-Wilk test and variance homogeneity with F-test. If both criteria were met, we applied the t-test; otherwise, we applied the Wilcoxon rank sum test. Both tests were set as two-tailed. In case of three-group comparisons, we implemented the Kruskal-Wallis Rank Sum test followed by Dunn's post hoc test. We applied correction for multiple comparisons inside each test hypothesis using Benjamini-Hochberg procedure.

The median fold change between groups was calculated for each comparison to extract the variables with the highest clinical usability. Only variables with a median fold change greater than 1.5 (either up- or downregulated) between tested groups and p-value FDR lower than 0.05 were considered significant and presented in the results section. Due to differences in valid values for each individual analyte tested, sample sizes (n) changed depending on missingness.

All calculations were performed using R (v. 4.1.0), with the packages tidyverse (general analysis), stats (statistical tests and hierarchical clustering), gplots (for heatmaps preparation), factoextra (for PCA analysis), ggsci (Figure color palette), ggdist (raincloud plots).

Pathway enrichment analysis was performed in R using packages biomaRt[47,48], ReactomePA[49] and clusterProfiler[50,51]. In case of metabolites, the relationship between metabolites and pathways was obtained from the Reactome website[52] and pathway enrichment was assessed by performing a hypergeometric test.

Classifier analysis was performed as follows. Initial on-study subject samples (median age: 4.7 weeks, Q1-Q3: 2.1–8.0 weeks) were analyzed by multivariable analysis. Epilepsy status was assessed at age 2 years, the end of the study. Patients who developed clinical or subclinical seizures by age 2 years had status of seizures present, else their status was seizure-free.

Seven subjects were excluded from this analysis due to occurrence of subclinical seizures at the time of study entry. 14 additional subjects were excluded because they had received Vigabatrin on a presymptomatic basis, and never developed seizures. Additionally, for 7 patients no comparable sample was available for inclusion. Hence, a total of 65 samples were analyzed, 54 from patients who developed seizures during the 2 year follow-up period, and 11 who did not. Sample size fluctuated due to missing data. Multiple data types were included in this analysis: Proteomics, metabolomics, transcriptomics, miRNA data, genotype information for 86 SNPs (Supplementary Data 1), TSC mutation status, and clinical data.

The 86 SNPs were selected based on review of published genetic association studies (review performed using PubMed literature database in April to May 2019). Selected SNPs had to have an allele associated with predisposition to epilepsy of any kind: focal or generalized epilepsy (including absence and juvenile myoclonic epilepsy), infantile spasms, or febrile and other types of seizures (Supplementary Data 11). The association finding had to be made in: (i) at least one genome-wide association study (GWAS), genome-wide meta-analysis association analysis, genome-wide mega-analysis association study, or candidate SNPs meta-analysis association study; or (ii) at least two independent candidate SNP association studies (Supplementary Data 11). SNPs were selected from studies performed in populations of any ethnicity, with no limits on population allele frequency or predicted risk effect, but had to be statistically significant. SNP genotypes were extracted from the EPISTOP WGS data[9], and were classified into two categories, either (i) major allele homozygote or (ii) heterozygote or minor allele homozygote.

For the classifier analysis each dataset was processed in the following way: first, constant features (columns with constant values spread across the whole dataset, or, with variance equal to zero) were excluded. Next, each variable was standardized (transformed to common scale), so the final values in each column had zero-mean and unit variance. More extensive filtering was used for the RNA-Seq dataset, since there were ~63,000 initial values. In the first filtering step, only genes with more than three reads in at least 75% of samples were retained. In the second step, a 2-fold median change between the seizure and no-seizure groups was required. Together, these filters reduced the number of RNA species under consideration to 267.

Analytes were selected for inclusion in the classifier analysis using univariate Area-under-the curve (AUC) metric to assess the ability of each variable to predict seizure appearance. Analytes had to meet an AUC threshold > 0.6 to be retained. In addition, no more than 30 analytes of each type could be selected, and they were down-sampled for inclusion based on AUC rank. This led to retention of 126 variables for inclusion in the classification algorithm from all analyte categories.

We used a logistic regression binary classification algorithm to generate a three-variable predictor of seizure development.

$$\ln\left(\frac{p}{1-p}\right) = \beta_0 + \beta_1 x_1 + \beta_2 x_2 + \beta_3 x_3 \qquad (3)$$

Where:

p – expected probability,

$x_{1,...,3}$ – independent variables,

$\beta_{0,...,3}$ – regression coefficients.

Logistic regression models the probability of an event (here: seizure development). If the calculated probability is higher than a defined threshold value, then the classifier predicts that patient will develop seizures (positive class). Otherwise the algorithm predicts that there will be no seizures (negative class). The probability threshold between positive and negative class was adjusted for each classifier separately due to high size imbalance between the classes.

The 126 analyte variables were combined into 1-, 2- and 3-predictor models. Over 330 thousand combinations were assessed for their association with seizure occurrence.

To validate our findings, we used the Monte-Carlo cross-validation method (repeated random subsampling). Each model was scored 100 times with the split of 2/3 in the training set and 1/3 in the test set with stratification to maintain the positive/negative case ratio in both sets. The goodness of fit with observed outcome was evaluated with Matthews Correlation Coefficient[19]:

$$\text{MCC} = \text{TP} \cdot \text{TN} - \text{FP} \cdot \text{FN}(\text{TP} + \text{FP}) \cdot (\text{TP} + \text{FN}) \cdot (\text{TN} + \text{FP}) \cdot (\text{TN} + \text{FN}),$$

$$(4)$$

Where:

MCC - Matthews Correlation Coefficient,

TP – true positives; samples correctly classified as positive (patients with seizures classified correctly),

TN – true negatives; samples correctly classified as negative (patients with seizures labelled as no seizures),

FP – false positives; samples incorrectly classified as positive (patients without seizures classified as with seizures),

FN - false negatives; samples incorrectly classified as negative (patients without seizures classified correctly).

Matthews Correlation Coefficient ranges from −1 (complete disagreement between prediction and observation) to 1 (complete agreement), with 0 meaning lack of correlation.

As the final metric for each classifier we chose the mean test MCC value obtained from Monte-Carlo cross-validation technique. Additionally, for each model, we present mean misclassification error (MMCE), positive predictive value (PPV) and negative predictive value (NPV). Missing values in the negative predictive value are related to the low number of negative examples in the test dataset. Due to the limited number of negative examples ($n_{max}$ for no seizures group = 12), the classifier sometimes incorrectly labeled all cases as positives, which caused the lack of true/false negatives (patients without seizures) in the formula for NPV calculation.

Due to the lack of an independent dataset for validation, we validated our results with a permutation test. To calculate the sampling distribution under the null hypothesis, we permuted the outcome variable (seizure status) and repeated the whole experiment 30 times – starting from feature selection to a model evaluation with subsampling (subsampling was done 50 times for each model to reduce the computation time). This resulted in about 16 million models. The test-statistic was set to the mean test MCC. The significance level was set at 0.05. A p value represents the fraction of models with random treatment allocation where the mean test MCC was equal or higher than in the models created for original data. Based on the permutation results, the calculated test statistics for this experiment had to exceed the mean MCC test equal to 0.73 to be considered significant.

The volume of data collected in the EPISTOP project consisted of about 33 TB. The experiment was designed in an R environment (version 4.1.0) using tidyverse, mlr, parallel and plotly packages (among all). We used packrat for package version control. The dedicated computing server consisted of 40 cores, 216 GB RAM and an additional 216 TB for data storage. Complete code for the

project is stored in the Github repository (https://github.com/JagGlo/molecular_EPISTOP).

### Reporting summary

Further information on research design is available in the Nature Portfolio Reporting Summary linked to this article.

## Data availability

The raw RNA-Seq data generated for this study are held at the European Genome-phenome Archive (EGA) under the accession number: EGAS00001007264. Access to this data is controlled by a data access committee. Please email JDM at james.mills@ucl.ac.uk or DJK at dk@rics.bwh.harvard.edu for further information. All other large data files have been combined into supplemental Data files S1A and S1B, which has been placed in our github repository along with the code for doing this analysis (https://github.com/JagGlo/molecular_EPISTOP), which has https://doi.org/10.5281/zenodo.8389826[53]. Source data are provided with this paper.

## Code availability

The code developed for and utilized in this study is openly available, and can be accessed at the Github repository (https://github.com/JagGlo/molecular_EPISTOP; https://doi.org/10.5281/zenodo.8389826[53]).

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

## Acknowledgements
The authors wish to thank all of the numerous support personnel who enabled enrollment of subjects, collection of samples, etc. for EPISTOP, and the infants and parents who participated in EPISTOP. EPISTOP was supported by the European Community's Seventh Framework Programme (FP7/2007-2013; EPISTOP, grant agreement no. 602391, SJ), the Tuberous Sclerosis Alliance (no. 02-13, DJK), and Polish Ministerial funds for science (years 2013-2019).

## Author contributions
D.K., E.A., J.J., B.J., K.Ko. and S.Jo. designed the study. K.Ko., A.J., B.P., B.W., C.H., D.D., F.J., J.B., K.R., K.Sa., L.L., M.F., P.C., P.K., R.M., R.N. and T.S. recruited and treated subjects, and collected clinical data. J.J., A.T., K.La. and M.U. processed clinical samples and distributed them to laboratories for analysis. J.J., K.La., M.U., B.J., D.K., E.A., A.I., A.M., C.M., F.H., J.J.A., J.M.A., J.D.M., J.vS., K.Le., and S.Ja. processed samples and performed technical analyses. K.La., D.K., F.H., J.D.M., J.vS., K.Le., J.G.W., J.Z., K.Kl., K.Si. and K.W. performed data and statistical analyses. D.K., F.H., J.D.M. and J.G.W. wrote the first draft of the manuscript. All authors read and contributed to the manuscript.

## Competing interests
The authors declare no competing interests.

## Additional information

Franz Huschner[1,30], Jagoda Głowacka-Walas ®[2,3,30], James D. Mills[4,5,6], Katarzyna Klonowska[7], Kathryn Lasseter[7], John M. Asara[8], Romina Moavero[9,10], Christoph Hertzberg[11], Bernhard Weschke ®[12], Kate Riney[13,14], Martha Feucht[15], Theresa Scholl[15], Pavel Krsek[16], Rima Nabbout[17], Anna C. Jansen[18], Bořivoj Petrák[16], Jackelien van Scheppingen[4],

Josef Zamecnik[19], Anand Iyer [20], Jasper J. Anink[4], Angelika Mühlebner[4,21], Caroline Mijnsbergen[4], Lieven Lagae[22], Paolo Curatolo[9], Julita Borkowska[23], Krzysztof Sadowski[23], Dorota Domańska-Pakieła[23], Magdalena Blazejczyk[23], Floor E. Jansen[24], Stef Janson[25], Malgorzata Urbanska [23], Aleksandra Tempes[26], Bart Janssen[25], Kamil Sijko [2], Konrad Wojdan[2,27], Sergiusz Jozwiak [23,28], Katarzyna Kotulska [23], Karola Lehmann[1], Eleonora Aronica[4,29], Jacek Jaworski[26] & David J. Kwiatkowski [7] ✉

[1]Proteome Factory AG, Berlin, Germany. [2]Transition Technologies Science, Warsaw, Poland. [3]Warsaw University of Technology, The Institute of Computer Science, Warsaw, Poland. [4]Amsterdam UMC, University of Amsterdam, Department of (Neuro)Pathology, Amsterdam Neuroscience, Amsterdam, The Netherlands. [5]Department of Clinical and Experimental Epilepsy, UCL Queen Square Institute of Neurology, London, UK. [6]Chalfont Centre for Epilepsy, Chalfont St Peter, UK. [7]Department of Medicine, Brigham and Women's Hospital, Boston, MA, USA. [8]Department of Medicine, Harvard Medical School and Division of Signal Transduction, Beth Israel Deaconess Medical Center, Boston, MA, USA. [9]Child Neurology and Psychiatry Unit, Systems Medicine Department, Tor Vergata University, Rome, Italy. [10]Developmental Neurology, Bambino Gesù Children's Hospital, IRCCS, Rome, Italy. [11]Diagnose- und Behandlungszentrum für Kinder, Vivantes-Klinikum Neukölln, Berlin, Germany. [12]Department of Child Neurology, Charité University Medicine Berlin, Berlin, Germany. [13]Neurosciences Unit, Queensland Children's Hospital, South Brisbane, Queensland, Australia. [14]School of Medicine, University of Queensland, St Lucia, Queensland, Australia. [15]Epilepsy Service, Department of Pediatrics and Adolescent Medicine, Medical University of Vienna, Member of ERN EpiCARE, Vienna, Austria. [16]Department of Paediatric Neurology, Motol University Hospital, 2nd Medical Faculty, Charles University, Prague, Czech Republic. [17]Department of Pediatric Neurology, Reference Centre for Rare Epilepsies, Necker–Enfants Malades Hospital, Université Paris cité, Imagine Institute, Paris, France. [18]Neurogenetics Research Group, Vrije Universiteit Brussel, Brussels, Belgium. [19]Department. of Pathology and Molecular Medicine, Motol University Hospital, 2nd Medical Faculty, Charles University, Prague, Czech Republic. [20]Department of Internal Medicine, Erasmus MC, Rotterdam, Netherlands. [21]Department of Pathology, University Medical Center Utrecht, Utrecht, The Netherlands. [22]Department of Development and Regeneration Section Pediatric Neurology, University Hospitals KU Leuven, Leuven, Belgium. [23]Department of Neurology and Epileptology, member of ERN EPICARE, The Children's Memorial Health Institute, Warsaw, Poland. [24]Department of Child Neurology, Brain Center University Medical Center Utrecht, Utrecht, The Netherlands. [25]GenomeScan, Leiden, The Netherlands. [26]International Institute of Molecular and Cell Biology, Warsaw, Poland. [27]Warsaw University of Technology, Institute of Heat Engineering, Warsaw, Poland. [28]Department of Child Neurology, Medical University of Warsaw, Warsaw, Poland. [29]Stichting Epilepsie Instellingen Nederland (SEIN), Heemstede the Netherlands, Utrecht, The Netherlands. [30]These authors contributed equally: Franz Huschner, Jagoda Głowacka-Walas. ✉e-mail: dk@rics.bwh.harvard.edu

