## [Peer Review File · Nature Communications]

Molecular EPISTOP, a comprehensive multi-omic analysis of blood from Tuberous Sclerosis Complex infants age birth to two yearsREVIEWER COMMENTS

Reviewer #1 (Remarks to the Author):

The present study that include extensive work from many groups reports important findings even though sometimes the findings are negative. The study was a tour de force that require significant effort from the authors and the patients. The data provide a reference point for future work. The authors identified key molecules that may be predictive of seizures in the TSC patient population. This will need to be further validated in both individuals with TSC and idiopathic epilepsy.

I am particularly excited about the data regarding:

miR-130a-3p, CECR7 and RADX , and I am looking forward to a followup study regarding the role of these molecules in seizures.

Reviewer #2 (Remarks to the Author):

“Molecular EPISTOP: Comprehensive multi-omic analysis of blood from Tuberous Sclerosis Complex infants age birth to two years” by Huschner et al.

Well written manuscript detailing findings from EpiSTOP, an important study looking at early intervention with vigabatrin in children with TSC. While seemingly done well with unbiased approach to identifying protein, RNA, and metabolomic differences in various groups with in EpiSTOP, no dramatic differences were identified within important groups:

“(II) Comparison of samples drawn at the time of, or shortly after, first seizure occurrence (nmax =86) versus samples from seizure free individuals (nmax = 95) showed no significant differences in protein groups, metabolites, or RNAs.” and

“(III) Similarly, comparison of samples drawn at the time of abnormal EEG detection (nmax = 45) versus those from subjects with normal EEG (nmax = 65) showed no significant differences in any analyte.”

This underscores the largest limitation of this study, that blood biomarkers reflect CNS dysfunction as related to TSC.

This concern is also borne out by data in Figure 4 showing that only one protein group, Collagen alpha-1(XI) chain was significantly higher in those with drug-resistant epilepsy and only hydroxyphenylacetic acid was significantly lower. Both of these findings were statistically significant but with non-overlapping values neither could distinguish a sample from an individual with drug-resistant epilepsy vs. seizure-free or drug controlled.”

Most interesting results are in Figure 6 with three- and two-variable models being reasonably predictive of seizure freedom.

Data in Figure 1G batch and VGB corrected doesn't seem to support the statement (line 408) that “dCMP showed a huge 52-fold increase in EPISTOP subjects on VGB”

The final sentence of the manuscript is apt: “Hence we view all of these observations as exploratory and tentative, and hope that a replication population may be studied for validation in the near future.”

Not sure what Fig. 3 adds to the manuscript and why it is only mentioned out of order in the Discussion. Maybe better to be Supplemental Figure?

All Tables are somewhat tedious to read and should be Supplemental.

REVIEWER COMMENTS

Reviewer #1 (Remarks to the Author):

The present study that includes extensive work from many groups reports important findings even though sometimes the findings are negative. The study was a tour de force that requires significant effort from the authors and the patients. The data provide a reference point for future work. The authors identified key molecules that may be predictive of seizures in the TSC patient population. This will need to be further validated in both individuals with TSC and idiopathic epilepsy.

I am particularly excited about the data regarding: miR-130a-3p, CECR7 and RADX, and I am looking forward to a followup study regarding the role of these molecules in seizures.

We are also excited about the data, of course. We do look forward to a followup/replication study for all of our findings, but this is beyond the scope of what we can do at present. We do not have the samples.

Reviewer #2 (Remarks to the Author):

“Molecular EPISTOP: Comprehensive multi-omic analysis of blood from Tuberous Sclerosis Complex infants age birth to two years” by Huschner et al.

Well written manuscript detailing findings from EpiSTOP, an important study looking at early intervention with vigabatrin in children with TSC. While seemingly done well with unbiased approach to identifying protein, RNA, and metabolomic differences in various groups with in EpiSTOP, no dramatic differences were identified within important groups:

“(II) Comparison of samples drawn at the time of, or shortly after, first seizure occurrence (nmax = 86) versus samples from seizure free individuals (nmax = 95) showed no significant differences in protein groups, metabolites, or RNAs.” and

“(III) Similarly, comparison of samples drawn at the time of abnormal EEG detection (nmax = 45) versus those from subjects with normal EEG (nmax = 65) showed no significant differences in any analyte.”

This underscores the largest limitation of this study, that blood biomarkers reflect CNS dysfunction as related to TSC.

This concern is also borne out by data in Figure 4 showing that only one protein group, Collagen alpha-1(XI) chain was significantly higher in those with drug-resistant epilepsy and only hydroxyphenylacetic acid was significantly lower. Both of these findings were statistically significant but with non-overlapping values neither could distinguish a sample from an individual with drug-resistant epilepsy vs. seizure-free or drug controlled.”

Yes, we were completely honest and straightforward with our findings. No change necessary.

Most interesting results are in Figure 6 with three- and two-variable models being reasonably predictive of seizure freedom.

We agree that these are interesting. However, validation is required.

Data in Figure 1G batch and VGB corrected doesn't seem to support the statement (line 408)

that “dCMP showed a huge 52-fold increase in EPISTOP subjects on VGB”

Note that there is a log 2 scale on the y axis as indicated. On the left in Figure 1G, batch-correction only has been performed; the median value for non-VGB subjects is about 13; the median value for the VGB-treated subjects is about 19. $19 - 13 = 6$, which on a log 2 scale converts to 64. Hence the reported (precise) difference of 52-fold is correct. On the right in Figure 1G, both batch and VGB exposure correction has been done, so there is no longer any difference between no VGB and VGB exposure.

The final sentence of the manuscript is apt: “Hence we view all of these observations as exploratory and tentative, and hope that a replication population may be studied for validation in the near future.”

No comment needed.

Not sure what Fig. 3 adds to the manuscript and why it is only mentioned out of order in the Discussion. Maybe better to be Supplemental Figure?

We have moved Figure 3 to the Supplement, and re-numbered the figures and S figures accordingly.

All Tables are somewhat tedious to read and should be Supplemental.

We agree that Table 1 is lengthy and should be moved to the supplement. However, Tables 2, 3, and 4 are important in providing the names of some analytes (proteins, metabolites, and RNAs) that were significant in different analyses, and we think may be of interest to the reader. This is recognizing that most readers do not access and read the supplemental material, and hence such information is not seen by those readers.

REVIEWER COMMENTS

Reviewer #1 (Remarks to the Author):

No more comments

Reviewer #2 (Remarks to the Author):

My concerns have been addressed satisfactorily and I appreciate the revisions made.

Reviewer #3 (Remarks to the Author):

- it's not clear at quick glance how the principle component analysis is set up, but the dimensions/components they are showing for RNA don't seem to describe much variance.
- authors discuss cibersort, but have no detail or data in paper.
- ambiguous language around how the RNA reads are filtered. Line 736 talks about one type of filtering and line 153 another. It's possible that line 153 happened first, but then the language around 736/153 should be reflective of what was done.
- RNA clustering was mentioned, it wasn't clear to me how that was done. Authors did provide a GitHub page.

Response to review of NCOMMS-23-08266A

Reviewer #3 comments:

Comment 1: it's not clear at quick glance how the principle component analysis is set up, but the dimensions/components they are showing for RNA don't seem to describe much variance.

Reply:

The PCA was performed using the R package factextra (as stated in the Methods lines 700-702 of previous version). We chose to show the plot of PC3 and PC4 for the RNA data since those two showed some correlation with age (Figure 2c). We did not show a plot of PC1 and PC2 (shown below) because neither showed correlation with age or any other clinical value. However, those two components account for a much larger share of variance (40.3%), as expected for the first two components. We could add this figure to the supplemental figures if desired by the reviewer/editor. No change was made to the manuscript.

Comment 2: authors discuss Cibersort, but have no detail or data in paper.

Reply:

We thank the reviewer for noticing this omission. We have now included the methods for CIBERSORT in the paper. The following text has now been added to the methods:

d. Deconvolution of bulk RNA-sequencing data

Deconvolution of the bulk RNA-sequencing was performed using CIBERSORTx¹⁷. The CIBERSORTx module “Cell Fractions” was used to enumerate the proportions of distinct cell subpopulations in the bulk RNA-Seq profile. This is performed by providing a matrix that has signature genes derived from either single cell transcriptomes or sorted cell populations. For the deconvolution of the EPISTOP RNA-Seq the publicly available LM22 signature matrix was used. LM22 is a signature genes file consisting of 547 genes that accurately distinguish 22 mature human hematopoietic populations and activation states, including seven T cell types, naïve and memory B cells, plasma cells, NK cells, and myeloid subsets.

Comment 3: ambiguous language around how the RNA reads are filtered. Line 736 talks about one type of filtering and line 153 another. It’s possible that line 153 happened first, but then the language around 736/153 should be reflective of what was done.

Reply:

The filtering methods applied to the RNA data may indeed be confusing for the reader and we thank the reviewer for pointing out this concern. In fact, different filters were used for the univariate analysis (line 153) and the classifier analysis (line 736). The filters described in lines 153 and 736 were done separately, each starting from the original read count table. For the univariate analysis (line 153), we used a more relaxed filter, retaining many more input RNAs (n=20,579), due to the nature of the analysis and statistical tests performed. For the classifier analysis (line 736), stricter filters were applied to lower the number of input RNA variables to 267, since this was a combinatorial analysis of 3 analytes, and could not be performed with a large number of RNA species.

To clarify this, we made the following changes to the text, shown in yellow here:

(previous lines 151-157)

481 protein groups were identified in the proteomics analysis (at false discovery rate (FDR) < 1%). 63,677 RNAs were identified by RNA-Seq (**Supplementary Table 1**). **For the univariate analyses**, RNAs and metabolites were retained for further analysis only if non-zero values were seen in $\geq 50\%$ samples; protein groups if seen in $\geq 70\%$. 20579 RNAs (13854 protein coding genes, 3221 long non-coding, 2598 of other biotypes including miRNAs, snRNAs,

pseudogenes, etc. and 906 RNAs that could not be assigned to any biotype), 249 metabolites, and 340 protein groups were retained for further analysis.

(733-740)

For the classifier analysis each dataset was processed in the following way: first, constant features (columns with constant values spread across the whole dataset, or, with variance equal to zero) were excluded. Next, each variable was standardized (transformed to common scale), so the final values in each column had zero-mean and unit-variance. More extensive filtering was used for the RNA-Seq dataset, since there were ~63,000 initial values. In the first filtering step, only genes with more than three reads in at least 75% of samples were retained. In the second step, a 2-fold median change between the seizure and no-seizure groups was required. Together, these filters reduced the number of RNA species under consideration to 267.

Comment 4: RNA clustering was mentioned, it wasn't clear to me how that was done. Authors did provide a GitHub page.

Reply:

We agree with the reviewer that it is not clear how the clustering of the RNA molecules was performed. Clustering of the RNA molecules was performed by first taking the median Z-score of each RNA across the different age bins (group median Z-score). The group median Z-scores for each RNA were then clustered using hierarchical clustering, using complete linkage in a dissimilarity matrix calculated using the Euclidean distance. This is now stated in the methods of the manuscript:

8. Developmental analysis

The assessment of developmental patterns in analytes involved several steps. The first step was to apply the Kruskal-Wallis Rank Sum test, alongside Dunn's post hoc test and Benjamini-Hochberg correction for multiple comparisons for each analyte and age category. Then for analytes with statistically significant differences among age groups, normalization was performed on all values for each molecule using Z-score standardization. To group molecules demonstrating similar patterns, the medians of standardized values were used as input for hierarchical clustering analysis. The process of hierarchical clustering using Complete linkage method was executed on a dissimilarity matrix derived from Euclidean distance calculations.

In addition, to make it clear which RNAs were subject to the clustering analysis (only those that were found to be differentially expressed with age, Kruskal Wallis: FDR <0.05), we also added the following line to the manuscript (changes in yellow)

*Hierarchical clustering of **age** group median Z-scores **that were found to be differentially expressed with age in the Kruskal-Wallis analysis (FDR < 0.05)** was used to identify similar patterns of expression changes among protein groups, metabolites, and RNA species (**Supplementary Fig. 3a-c**). Based on visual inspection and tuning, different clusters with similar changes in levels with age were defined for protein groups (6 clusters), metabolites (6), and RNAs (8), respectively (**Supplementary Fig. 3a-c, Supplementary Table 3**).*